# Precursor-Directed Biosynthesis Mediated Amplification of Minor Aza Phenylpropanoid Piperazines in an Australian Marine Fish-Gut-Derived Fungus, *Chrysosporium* sp. CMB-F214

**DOI:** 10.3390/md19090478

**Published:** 2021-08-25

**Authors:** Ahmed H. Elbanna, Amila Agampodi Dewa, Zeinab G. Khalil, Robert J. Capon

**Affiliations:** 1Institute for Molecular Bioscience, The University of Queensland, St Lucia, QLD 4072, Australia; ahmed.elbanna@pharma.cu.edu.eg (A.H.E.); a.agampodidewa@imb.uq.edu.au (A.A.D.); z.khalil@uq.edu.au (Z.G.K.); 2Department of Pharmacognosy, Faculty of Pharmacy, Cairo University, Cairo 11562, Egypt

**Keywords:** marine fish-gut-derived fungus, *Chrysosporium* sp., phenylpropanoid piperazines, chrysosporazines, azachrysosporazines, spirochrysosporazine, precursor-directed biosynthesis, P-glycoprotein inhibition, multidrug resistance

## Abstract

Chemical analysis of an M1 agar plate cultivation of a marine fish-gut-derived fungus, *Chrysosporium* sp. CMB-F214, revealed the known chrysosporazines A–D (**11**–**14**) in addition to a suite of very minor *aza* analogues **1**–**6**. A microbioreactor (MATRIX) cultivation profiling analysis failed to deliver cultivation conditions that significantly improved the yields of **1**–**6**; however, it did reveal that M2 agar cultivation produced the new natural product **15**. A precursor-directed biosynthesis strategy adopting supplementation of a CMB-F214 M1 solid agar culture with sodium nicotinate enhanced production of otherwise inaccessible azachrysposorazines A1 (**1**), A2 (**2**), B1 (**3**), C1 (**4**), C2 (**5**) and D1 (**6**), in addition to four new chrysosporazines; chrysosporazines N–P (**7**–**9**) and spirochrysosporazine A (**10**). Structures inclusive of absolute configurations were assigned to **1**–**15** based on detailed spectroscopic and chemical analyses, and biosynthetic considerations. Non-cytotoxic to human carcinoma cells, azachrysosporazies **1**–**5** were capable of reversing doxorubicin resistance in P-glycoprotein (P-gp)-overexpressing human colon carcinoma cells (SW620 Ad300), with optimum activity exhibited by the C-2′ substituted analogues **3**–**5**.

## 1. Introduction

During prior investigations into Australian marine-derived fungi, we reported on the gastrointestinal tract (GIT) of fresh market-purchased fish (Mugil mullet) as a rich source of taxonomically and chemically diverse fungi. We went on to report on the discovery of rare lipodepsipeptide scopularides from *Scopulariopsis* spp. CMB-F458 and CMB-F115, and *Beauveria* sp. CMB-F585 [1]; unprecedented hydrazine *N*-amino-l-proline methyl ester and associated Schiff base artifact prolinimines from *Evlachovaea* sp. CMB-F563 [2,3]; and *N*-benzoyl and *N*-cinnamoyl phenylpropanoid piperazine chrysosporazines from *Chrysosporium* spp. CMB-F214 and CMB-F294, respectively [4,5]. The chrysosporazines were particular noteworthy, being non-cytotoxic to human carcinoma cells but exhibiting promising inhibitory activity against the multidrug resistance efflux pump P-glycoprotein (P-gp). For example, P-gp-overexpressing human colon carcinoma (SW620 Ad300) cells pre-treated with chrysosporazine F (2.5 µM) acquired a gain in sensitivity (GS 14) against the anticancer agent doxorubicin, >2-fold that of the positive control verapamil (GS 6.1), making chrysosporazine F one of the more potent P-gp inhibitors reported to date [5]. We now report a precursor-directed biosynthesis strategy where supplementation of a CMB-F214 M1 solid agar culture with sodium nicotinate enhanced production of otherwise inaccessible new natural products—namely azachrysposorazines A1 (**1**), A2 (**2**), B1 (**3**), C1 (**4**), C2 (**5**) and D1 (**6**), chrysosporazines N–P (**7**–**9**) and spirochrysosporazine A (**10**)—along with the known chrysosporazines A–D (**11**–**14**). By contrast, a CMB-F214 M2 solid agar culture without precursor supplementation produced a new natural product, chrysosporazine Q (**15**). Structures were assigned to **1**–**15** (Figure 1) on the basis of detailed spectroscopic and chemical analyses, and biosynthetic considerations. Access to **1**–**15** facilitated a structure–activity relationship analysis on the P-gp inhibitory properties of this novel class of phenylpropanoid piperazines.

## 2. Results and Discussion

UPLC-DAD (210 nm) analysis of an M1 agar plate cultivation of CMB-F214 revealed the known chrysosporazines A–D (**11**–**14**) accompanied by a suite of earlier-eluting very minor co-metabolites **1**–**6** (Figure 2A). A microbioreactor (MATRIX) cultivation profiling analysis employing 12 media compositions under solid phase as well as shaken and static broth failed to deliver cultivation conditions that significantly improved the yields of **1**–**6**; however, it did reveal that M2 solid-phase agar cultivations produced the new natural product **15** (Figure 2B and Appendix A). Notwithstanding low yields for **1–6**, a trace amount of pure **4** recovered from a large-scale CMB-F214 rice cultivation possessed a molecular formula (C_28_H_27_N_3_O_5_) suggestive of an *aza* analogue of the co-metabolite chrysosporazine C (**13**) (C_29_H_28_N_2_O_5_). Consistent with this hypothesis, the (albeit limited) 1D NMR (DMSO-*d*_6_) data for **4** indicated incorporation of an *N*-nicotinoyl moiety, rather than the *N*-benzoyl moiety found in **13**, leading to speculation that nicotinic acid (i.e., niacin, vitamin B_3_) may be a rate-limiting biosynthetic precursor. This view was validated when analytical scale M1 solid-phase and static broth cultures of CMB-F214 supplemented with sodium nicotinate generated enhanced yields of the target *aza* analogues **1**–**6** (Figure 2A and Appendix A), as well as four new exemplars (**7**–**10**) of the chrysosporazine class. A large-scale 18-day cultivation of CMB-F214 on M1 agar plates supplemented with sodium nicotinate (2 mg/mL) was subjected to solvent extraction, trituration and reversed phase fractionation to yield the new metabolites **1**–**10**, while comparable treatment of a large-scale 8-day M2 agar plate cultivation yielded the new metabolite **15**. An account of the structure elucidation of **1**–**10** and **15** is outlined below.

HRESI(+)MS measurements on **1** and **2** returned isomeric molecular formulae (C_21_H_19_N_3_O_5_, Δmmu +0.2 and +0.7, respectively) consistent with *aza* analogues of chrysosporazine A (**11**) (C_22_H_20_N_2_O_5_). Comparison of the 1D NMR (DMSO-*d*_6_) data for **1** (Table 1 and Table 2, Appendix A, Appendix A) with **11** revealed many similarities, with key differences attributed to replacement of the C-3″ to C-3 cyclised *N*-benzoyl moiety in **11**, with a C-3″ to C-3 cyclised *N*-nicotinoyl moiety in **1**. The N-4″ regiochemistry in **1** was evident from the contiguous nature and significant deshielding of H-5″ (δ_H_ 8.56, dd, 4.8, 1.8 Hz) and H-7″ (δ_H_ 8.26, dd, 7.8, 1.8 Hz) relative to H-6″ (δ_H_ 7.43, ddd, 7.8, 4.8, 0.8 Hz), as well as diagnostic 2D NMR correlations (Figure 3). Comparison of the 1D NMR (DMSO-*d*_6_) data for **2** (Table 1 and Table 2, Appendix A, Appendix A) with that for **1** and **11** attributed differences to an alternate N-6″ *N*-nicotinoyl regiochemistry, as would be expected by cyclisation of an alternate C-1″ to C-2″ rotamer to C-3. The N-6″ regiochemistry was evident from *J*_4″,5″_ ortho coupling (5.0 Hz) and lack of coupling to H-7″, deshielding of H-5″ (δ_H_ 8.59) and H-7″ (δ_H_ 9.03) relative to H-4″ (δ_H_ 6.66), and diagnostic 2D NMR correlations (Figure 3). The *E* rotamer configuration of the acetamide in **1** and **2** was evident from ROESY correlations between N-COCH_3_ and H-2′ (Figure 3), while excellent concordance among key ^1^H NMR resonances in **1** (δ_H_ 4.47, ddd, 12.3, 9.8, 3.0 Hz, H-2; 4.51, d, 12.3 Hz, H-3) and **2** (δ_H_ 4.49, ddd, 12.1, 9.8, 3.0 Hz, H-2; 4.46, d, 12.1 Hz, H-3) with **11** (δ_H_ 4.44, ddd, 12.2, 10.4, 3.1 Hz, H-2; 4.36, d, 12.2 Hz, H-3) suggested a common diaxial relative configuration. These observations, together with biosynthetic considerations and the fact the absolute configuration for **11** had been prior confirmed by X-ray analysis [4] permitted assignment of the structures for azachrysosporazines A1 (**1**) and A2 (**2**) as shown.

HRESI(+)MS measurement on **3** returned a molecular formula (C_28_H_25_N_3_O_6_, Δmmu –0.5) consistent with an *aza* analogue of chrysosporazine B (**12**) (C_29_H_26_N_2_O_6_). Comparison of the 1D NMR (DMSO-*d*_6_) data for **3** (Table 1 and Table 2, Appendix A, Appendix A) with **12** revealed many similarities, including the presence of major and minor acetamide rotamers (ratio 1:0.3), with key differences attributed to replacement of the C-3″ to C-3 cyclised *N*-benzoyl moiety in **12**, with a C-3″ to C-3 cyclised *N*-nicotinoyl moiety in **3**. The N-4″ regiochemistry in **3** was evident from the contiguous nature and significant deshielding of H-5″ (δ_H_ 8.64, dd, 4.7, 1.8 Hz) and H-7″ (δ_H_ 8.18, dd, 7.8, 1.8 Hz) relative to H-6″ (δ_H_ 7.44, dd, 7.8, 4.7 Hz), as well as diagnostic 2D NMR correlations (Figure 4). ROESY correlations between N-COCH_3_ and Ha-1 in **3** permitted assignment of a *Z* configuration about the major acetamide rotamer (Figure 4), while excellent concordance among key ^1^H NMR resonances in **3** (δ_H_ 4.21, ddd, 9.8, 5.4, 3.8 Hz, H-2; 4.46, d, 5.4 Hz, H-3; 5.89, dd, 5.7, 1.3 Hz, H-2′) and **12** (δ_H_ 4.20, ddd, 8.5, 8.0, 3.9 Hz, H-2; 4.39, d, 8.0 Hz, H-3; 5.86, dd, 5.5, 1.3 Hz, H-2′) supported a common relative configuration. This, together with biosynthetic considerations and the fact the absolute configuration for **12** had been prior confirmed by X-ray analysis [4], permitted assignment of the structure for azachrysosporazine B1 (**3**) as shown.

HRESI(+)MS measurements on **4** and **5** returned isomeric molecular formulae (C_28_H_27_N_3_O_5_, Δmmu +1.2 and +1.2, respectively) consistent with *aza* analogues of chrysosporazine C (**13**) (C_29_H_28_N_2_O_5_). Comparison of the 1D NMR (DMSO-*d*_6_) data for **4** (Table 1 and Table 2, Appendix A, Appendix A) with **13** revealed many similarities, including the presence of major and minor acetamide rotamers (ratio 1:0.2), with key differences attributed to replacement of the C-3″ to C-3 cyclised *N*-benzoyl moiety in **13**, with a C-3″ to C-3 cyclised *N*-nicotinoyl moiety in **4**. The N-4″ regiochemistry in **4** was evident from the contiguous nature and significant deshielding of H-5″ (δ_H_ 8.60, dd, 4.7, 1.8 Hz) and H-7″ (δ_H_ 8.18, dd, 7.8, 1.8 Hz) relative to H-6″ (δ_H_ 7.45, dd, 7.8, 4.7 Hz), as well as diagnostic 2D NMR correlations (Figure 4). Comparison of the 1D NMR (DMSO-*d*_6_) data for **5** (Table 1 and Table 2, Appendix A, Appendix A) with that for **4** and **13** revealed many similarities, including the presence of major and minor acetamide rotamers (ratio 1: 0.2), with key differences attributed to an alternate N-6″ *N*-nicotinoyl regiochemistry, as would be expected by C-3 cyclisation with an alternate C-1″ to C-2″ rotamer. The N-6″ regiochemistry was evident from deshielding of H-5″ (δ_H_ 8.60) and H-7″ (δ_H_ 9.08) relative to H-4″ (δ_H_ 6.72), and diagnostic 2D NMR correlations (Figure 4). ROESY correlations between N-COCH_3_ and H-2′ in **4** and **5** (in common with **13**) permitted assignment of a *E* configuration for the major acetamide rotamers (Figure 4), with excellent concordance among key ^1^H NMR resonances in **4** (δ_H_ 3.88, ddd, 11.4, 10.0, 3.8 Hz, H-2; 4.49, d, 10.0 Hz, H-3; 4.56, dd, 13.4, 1.1 Hz, Ha-1′; 3.00, m, Hb-1′; 4.25, m, H-2′), **5** (δ_H_ 3.91, ddd, 11.2, 11.0, 3.9 Hz, H-2; 4.46, d, 11.2 Hz, H-3; 4.56, dd, 13.5, 1.2 Hz, Ha-1′; 2.95, dd, 13.5, 4.1 Hz, Hb-1′; 4.27, m, H-2′) and **13** (δ_H_ 3.85, ddd, 11.0, 11.0, 3.9 Hz, H-2; 4.38, d, 11.0 Hz, H-3; 4.58, dd, 13.5, 1.2 Hz, Ha-1′; 2.95, dd, 13.5, 4.2, Hb-1′; 4.25, m, H-2′) supporting a common relative configuration. This, together with biosynthetic considerations, permitted assignment of the structures for azachrysosporazines C1 (**4**) and C2 (**5**) as shown.

HRESI(+)MS measurement on **6** returned a molecular formula (C_28_H_29_N_3_O_5_, Δmmu 0.0) consistent with an *aza* analogue of chrysosporazine D (**14**) (C_29_H_30_N_2_O_5_). As with **14**, the NMR (DMSO-*d*_6_) data for **6** (Appendix A) was heavily broadened, suggestive of equilibrating rotamers. UPLC-DAD-QTOF analysis of the acid hydrolysate prepared from **6** yielded a peak (*m*/*z* [M + H]^+^ 341, C_20_H_24_N_2_O_3_) (Appendix A) that co-eluted with an authentic sample of the piperazine **16** (Figure 5) previously prepared and fully characterised following acid hydrolysis of **14** [4]. These observations, together with biosynthetic considerations, permitted assignment of the structure for azachrysosporazine D1 (**6**) as shown.

HRESI(+)MS measurements on **7** and **8** returned isomeric molecular formulae (C_29_H_28_N_2_O_6_, Δmmu −0.6 and −0.1, respectively) consistent with reduced (+H_2_) analogues of chrysosporazine B (**12**). Comparison of the 1D NMR (DMSO-*d*_6_) data for **7** and **8** (Table 3 and Table 4 and Appendix A, Appendix A) with **12** revealed many similarities, including major and minor acetamide rotamers (ratio 1:0.2 in **7**, 1:0.3 in **8**), with differences attributed to replacement of the asymmetric 6,7-methylenedioxy-8-methoxy-4-benzyl moiety in **12** with a symmetric 6,8-dimethoxy-7-hydroxy-4-benzyl moiety in **7** and **8**. Diagnostic 2D NMR correlations supported this structure, with a ROESY correlation between N-COCH_3_ and H-1 confirming the major acetamide rotamers in both **7** and **8** having a *Z* configuration (Figure 6). Excellent concordance in key ^1^H NMR resonances for **7** (δ_H_ 4.27, ddd, 12.4, 9.0, 4.1 Hz, H-2; 4.36, d, 9.0 Hz, H-3; 4.76, dd, 14.3, 1.9 Hz, Ha-1′; 3.52, dd, 14.3, 5.5, Hb-1′; 5.83, dd, 5.5, 1.9, H-2′) and **12** (δ_H_ 4.20, ddd, 8.5, 8.0, 3.9 Hz, H-2; 4.39, d, 8.0 Hz, H-3; 4.78, dd, 14.1, 1.3 Hz, Ha-1′; 3.48, dd, 14.1, 5.5, Hb-1′; 5.86, dd, 5.5, 1.3, H-2′) supported a common relative configuration, which together with biosynthetic considerations permitted assignment of the structure for chrysosporazine N (**7**) as shown. By contrast, key differences in the ^1^H NMR data for **8** (δ_H_ 4.37, dd, 14.1, 7.1 Hz, Ha-1′; 3.98, dd, 14.1, 4.5, Hb-1′; 5.67, dd, 7.1, 4.5, H-2′) compared to both **7** and **12**, together with biosynthetic considerations, supported assignment of the 2′-epimer structure for chrysosporazine O (**8**) as shown.

HRESI(+)MS measurement on **9** returned a molecular formula (C_29_H_32_N_2_O_5_, Δmmu +1.0) consistent with a reduced (+H_2_) analogue of chrysosporazine D (**14**). As with **14**, the NMR (DMSO-*d*_6_) data for **9** (Appendix A) was heavily broadened, and indicative of an equilibrating mixture of acetamide and benzamide rotamers. Fortuitously, cultivation of CMB-F214 on M2 agar plates yielded a new metabolite **15**, which HRESI(+)MS measurement attributed a molecular formula (C_22_H_28_N_2_O_4_, Δmmu 0.0). Analysis of the 1D NMR (DMSO-*d*_6_) data for **15** (Table 3 and Table 4 and Appendix A, Appendix A) revealed the now almost ubiquitous major and minor acetamide rotamers (ratio 1:0.3), a symmetric 6,8-dimethoxy-7-hydroxy-4-benzyl moiety in common with **7** and **8**, and a core piperazine bearing an unsubstituted benzyl moiety. Diagnostic 2D NMR correlations permitted assembly of the planar structure, with a ROESY correlation between N-COCH_3_ and H-2′ confirming an *E* configuration for the major acetamide rotamer (Figure 6). These observations, together with biosynthetic considerations and the fact that acid hydrolysis of **9** and **15** yielded a common co-eluting piperazine **19** (along with production of **15** from **9**) (Figure 5, Appendix A), allowed assignment of the structures for chrysosporazine P (**9**) and Q (**15**) as shown.

HRESI(+)MS measurement on **10** returned a molecular formula (C_20_H_22_N_2_O_5_, Δmmu +0.1) suggestive of truncated analogue of chrysosporazine Q (**15**). Analysis of the 1D NMR (DMSO-*d*_6_) data for **10** (Table 3 and Table 4 and Appendix A, Appendix A) revealed major and minor acetamide rotamers (ratio 1:0.6), with resonances attributed to C-1′ to C-9′ in common with **15**. Consideration of diagnostic 2D NMR correlations allowed assembly of the planar structure for **10**, with a ROESY correlation between N-COCH_3_ and H-2′ confirming an *E* configuration about the major acetamide rotamer (Figure 7). ROESY correlations between H-2 and H_α_-3, and between H_β_-3 and H-7, established the relative configuration about C-2/C-4, which together with biosynthetic considerations permitted assignment of the structure for spirochrysosporazine A (**10**) as shown. A plausible biosynthesis pathway leading to **10** could proceed via enzyme-mediated (stereospecific) oxidative aromatic ring contraction and sequential lactonisation and lactamisation of **15** (Figure 8).

As with chrysosporazines **11**–**14**, the new metabolites **1**–**10** and **15** did not show any growth inhibitory properties (IC_50_ > 30 µM) against the Gram-positive and Gram-negative bacteria, the fungus *Candida albicans* (Appendix A), or human colon (SW620) and P-glycoprotein (P-gp)-overexpressing human colon (SW620 Ad300) carcinoma cells (Figure 9). Significantly, **1**–**10** and **15** reversed doxorubicin resistance in SW620 Ad300 carcinoma cells, with **1** and **2** inducing a gain in sensitivity (GS) comparable to, and **3**–**5** >2.5-fold that of the positive control verapamil (Table 5, Figure 9). A structure–activity relationship analysis based on these results suggests the methylenedioxy ring, C-3/C-3″ cyclisation and C-2′ substitution are key determinants for improved P-gp inhibition.

## 3. Materials and Methods

### 3.1. Chrysosporium sp. CMB-F214 Collection, Isolation and Taxonomy

The fungus *Chrysosporium sp.* CMB-F214 was isolated from the gastrointestinal tract of a specimen of *Mugil* mullet fish, on an M1 agar plate in the presence of 3.3% artificial sea salt (M1S), and incubated at 26.5 °C for eight days. Genomic DNA for CMB-F214 was extracted from its mycelia using the DNeasy Plant Mini Kit (Qiagen, Brisbane, Australia) as per the manufacturer’s protocol and as previously described [4]. A BLAST analysis (NCBI database) on the amplified ITS gene sequence (GenBank accession no. MN249497) revealed 99% homology with *Chrysosporium lobatum*.

### 3.2. Chrysosporium sp. CMB-F214 Media MATRIX Study

From an agar plate culture of the fungus *Chrysosporium* sp. CMB-F214, spores were transferred into 24-well microbioreactors (MBRs) charged with a range of different culture media (×12), and in solid (2.5 mL), broth static (1.5 mL) and broth shaken (1.5 mL) formats. MBRs were incubated at 26.5 °C for 8 days, with 190 rpm for shaken broth. After incubation, individual wells were extracted in situ with EtOAc (2 mL, each), filtered and dried under N_2_. The resulting extracts were dissolved in MeOH (100 µL for solid and 50 µL for broth, each containing trace levels of an internal calibrant) and analysed by Ultra-High Performance Liquid Chromatography-diode array detector (UPLC-DAD) and Ultra-High Performance Liquid Chromatography-quadrupole time of flight (UPLC-QTOF), with the resulting chromatograms compared at the same scale (Appendix A)

### 3.3. Analytical Precursor-Directed (Nicotinate) Feeding Study

Sodium salts of nicotinic acid were prepared by dissolving in equal amounts of saturated NaHCO_3_ solution and sterile H_2_O to provide the required concentrations (pH ~7). A 24-well MBR was used to obtain miniaturised cultures where M1 broth media (1400 µL) was mixed with the corresponding sodium salt concentration (100 µL) to provide final concentrations of 2 and 4 mg/mL. A single colony of *Chrysosporium* sp. CMB-F214 was inoculated in each well and incubated under static conditions for eight days at 26.5 °C. After incubation, the cultures were extracted in situ with EtOAc (2 mL), filtered and dried under N_2_ affording the corresponding extracts. These extracts were dissolved in MeOH (100 µL) and subjected to UPLC-DAD and UPLC-QTOF analyses. For the agar plate cultures, a similar procedure was adopted. The sodium nicotinate solution was prepared, mixed with M1 agar media before solidification (2 mg/mL as final concentration), poured to plates and left to solidify. Spores of the fungus CMB-F214 were inoculated on the agar plates and incubated for eight days at 26.5 °C. After incubation, the agar cultures were chopped, extracted with EtOAc, filtered**,** dried under N_2_ and analysed by UPLC-DAD and UPLC-QTOF. (Figure 2A and Appendix A)

### 3.4. M2 Agar Scale-up Culture and Production of Chrysosporazine Q (**15**)

The fungus *Chrysosporium* sp. CMB-F214 was inoculated on M2 agar (×40 plates), incubated at 26.5 °C for eight days, after which the agar cultures were chopped, extracted with EtOAc (2 × 400 mL), filtered and concentrated in vacuo at 40 °C, to yield the crude extract (300 mg). The crude extract was then defatted using *n*-hexane (20 mL × 3) providing the defatted crude extract (250 mg). A portion of the latter (200 mg) was treated with MeOH (2 mL), sonicated, centrifuged and the supernatant was subjected to gel chromatography (Sephadex^®^ LH-20 2.5 cm × 70 cm column, gravity elution with isocratic MeOH). Fractions (10 mL) were collected and monitored by UPLC-DAD and UPLC-QTOF, and collective fraction A (75 mg) was rich in target compound **15**. An aliquot of fraction A (20 mg) was subjected to semi-preparative reversed phase HPLC (ZORBAX SB-C_8_ 9.4 mm × 25 cm, 5 µm, 3 mL/min gradient elution 90–0% H_2_O/MeCN inclusive of 0.1% TFA modifier over 20 min) to yield chrysosporazine Q (**15**) (7 mg). (Appendix A)

### 3.5. Chrysosporium sp. CMB-F214 Culture Supplemented with Sodium Nicotinate Leading to Amplified Production of the Minor Natural Products, Azachrysosporazines ***1**–**6***, New Chrysosporazines ***7**–**9*** and Spirochrysosporazine A (***10***)

Sodium nicotinate solution was prepared by dissolving nicotinic acid (3.2 g) in saturated NaHCO_3_ solution (26.6 mL) and H_2_O (26.6 mL) (pH~7). The salt solution was added to warm M1 agar media (final concentration of 2 mg/mL) and stirred for 5 min, then poured to plates and allowed to cool down. The fungus *Chrysosporium* sp. CMB-F214 was inoculated on the agar plates (×80), incubated at 26.5 °C for 18 days, then extracted with EtOAc (4 × 500 mL), filtered and concentrated in vacuo at 40 °C, to yield the crude extract (482 mg). The crude extract was then triturated with *n*-hexane (20 mL × 3) providing, after in vacuo concentration, the *n*-hexane solubles (98 mg) and the defatted MeOH solubles (380 mg). The MeOH solubles fraction (380 mg) was further fractionated by preparative reversed phase HPLC (Phenomenex Luna-C_8_ 21.2 mm × 25 cm × 10 µm, 20 mL/min 90–0% H_2_O/MeCN gradient elution over 20 min inclusive of an isocratic 0.1% TFA). Fractions A–E were determined as rich in target compounds (**1–10**) on the basis of UPLC-DAD and UPLC-QTOF analyses. Fraction A (4.2 mg) was subjected to a semi-preparative reversed phase HPLC (ZORBAX SB-C_3_ 9.4 mm × 25 cm, 5 µm, 3 mL/min isocratic elution 70% H_2_O/MeCN inclusive of 0.1% TFA modifier over 33 min) to yield azachrysosporazine A2 (**2**) (1.2 mg). Fraction B (5.5 mg) was subjected to a semi-preparative reversed phase HPLC (ZORBAX SB-C_3_ 9.4 mm × 25 cm, 5 µm, 3 mL/min isocratic elution 75% H_2_O/MeCN inclusive of 0.1% TFA modifier over 37 min) to yield azachrysosporazine A1 (**1**) (1.0 mg) and spirochrysosporazine A (**10**) (1.8 mg). Fraction C (5.0 mg) was subjected to a semi-preparative reversed phase HPLC (ZORBAX SB-C_3_ 9.4 mm × 25 cm, 5 µm, 3 mL/min isocratic elution 62% H_2_O/MeCN inclusive of 0.1% TFA modifier over 35 min) to yield azachrysosporazine C2 (**5**) (2.0 mg) and azachrysosporazine D1 (**6**) (0.4 mg). Fraction D (7.0 mg) was subjected to a semi-preparative reversed phase HPLC (ZORBAX SB-C_3_ 9.4 mm × 25 cm, 5 µm, 3 mL/min isocratic elution 62% H_2_O/MeCN inclusive of 0.1% TFA modifier over 25 min) to yield azachrysosporazine B1 (**3**) (1.4 mg), chrysosporazine N (**7**) (1.4 mg) and chrysosporazine O (**8**) (1.1 mg). Fraction E (5.5 mg) was subjected to a semi-preparative reversed phase HPLC (ZORBAX SB-C_3_ 9.4 mm × 25 cm, 5 µm, 3 mL/min isocratic elution 75% H_2_O/MeCN inclusive of 0.1% TFA modifier over 35 min) to yield azachrysosporazine C1 (**4**) (1.5 mg) and chrysosporazine P (**9**) (1.2 mg). (Appendix A).

### 3.6. Metabolites’ Characterisation


Azachrysosporazine A1 (**1**); yellow oil; [α]_D_^21.2^ − 88.4 (c 0.083, MeOH); NMR (600 MHz, DMSO-*d*_6_) see Table 1 and Table 2 and Appendix A, Appendix A; ESI(+)MS *m*/*z* 394 [M + H]+; HRESI(+)MS *m*/*z* 394.1399 [M + H]+ (calcd for C_21_H_20_N_3_O_5_, 394.1397).Azachrysosporazine A2 (**2**); yellow oil; [α]_D_^21.2^ − 55.0 (c 0.1, MeOH); NMR (600 MHz, DMSO-*d*_6_) see Table 1 and Table 2 and Appendix A, Appendix A; ESI(+)MS *m*/*z* 394 [M + H]+; HRESI(+)MS *m*/*z* 394.1404 [M + H]+ (calcd for C_21_H_20_N_3_O_5_, 394.1397).Azachrysosporazine B1 (**3**); yellow oil; [α]_D_^21.2^ + 43.1 (c 0.116, MeOH); NMR (600 MHz, DMSO-*d*_6_) see Table 1 and Table 2 and Appendix A, Appendix A; ESI(+)MS *m*/*z* 500 [M + H]+; HRESI(+)MS *m*/*z* 500.1831 [M + H]+ (calcd for C_28_H_26_N_3_O_6_, 500.1816).Azachrysosporazine C1 (**4**); yellow oil; [α]_D_^21.2^ − 29.6 (c 0.125, MeOH); NMR (600 MHz, DMSO-*d*_6_) see Table 1 and Table 2 and Appendix A, Appendix A; ESI(+)MS *m*/*z* 486 [M + H]+; HRESI(+)MS *m*/*z* 486.2035 [M + H]+ (calcd for C_28_H_28_N_3_O_5_, 486.2023).Azachrysosporazine C2 (**5**); yellow oil; [α]_D_^21.2^ − 36.3 (c 0.166, MeOH); NMR (600 MHz, DMSO-*d*_6_) see Table 1 and Table 2 and Appendix A, Appendix A; ESI(+)MS *m*/*z* 486 [M + H]+; HRESI(+)MS *m*/*z* 486.2035 [M + H]+ (calcd for C_28_H_28_N_3_O_5_, 486.2023).Azachrysosporazine D1 (**6**); yellow oil; [α]_D_^22.2^ + 19.3 (c 0.033, MeOH); NMR (600 MHz, DMSO-*d*_6_) see Appendix A; ESI(+)MS *m*/*z* 488 [M + H]+; HRESI(+)MS *m*/*z* 488.2180 [M + H]+ (calcd for C_28_H_30_N_3_O_5_, 488.2180).Chrysosporazine N (**7**); yellow oil; [α]_D_^21.2^ + 5.3 (c 0.166, MeOH); NMR (600 MHz, DMSO-*d*_6_) see Table 3 and Table 4 and Appendix A, Appendix A; ESI(+)MS *m*/*z* 501 [M + H]+; HRESI(+)MS *m*/*z* 501.2014 [M + H]+ (calcd for C_29_H_29_N_2_O_6_, 501.2020).Chrysosporazine O (**8**); yellow oil; [α]_D_^21.2^ − 86.8 (c 0.185, MeOH); NMR (600 MHz, DMSO-*d*_6_) see Table 3 and Table 4 and Appendix A, Appendix A; ESI(+)MS *m*/*z* 501 [M + H]+; HRESI(+)MS *m*/*z* 501.2019 [M + H]+ (calcd for C_29_H_29_N_2_O_6_, 501.2020).Chrysosporazine P (**9**); yellow oil; [α]_D_^22.2^ + 18.4 (c 0.08, MeOH); NMR (600 MHz, DMSO-*d*_6_) see Appendix A; ESI(+)MS *m*/*z* 489 [M + H]+; HRESI(+)MS *m*/*z* 489.2394 [M + H]+ (calcd for C_29_H_33_N_2_O_5_, 489.2384).Spirochrysosporazine A (**10**); yellow oil; [α]_D_^21.0^ − 76.6 (c 0.154, MeOH); NMR (600 MHz, DMSO-*d*_6_) see Table 3 and Table 4 and Appendix A, Appendix A; ESI(+)MS *m*/*z* 371 [M + H]+; HRESI(+)MS *m*/*z* 393.1422 [M + Na]+ (calcd for C_20_H_22_N_2_O_5_Na, 393.1421).Chrysosporazine Q (**15**); White powder; [α]_D_^22.5^ − 28.2 (c 0.286, MeOH); NMR (600 MHz, DMSO-*d*_6_) see Table 3 and Table 4 and Appendix A, Appendix A; ESI(+)MS *m*/*z* 385 [M + H]+; HRESI(+)MS *m*/*z* 385.2122 [M + H]+ (calcd for C_22_H_29_N_2_O_4_, 385.2122).


### 3.7. Acid Hydrolysis of Azachrysosporazine D1 (***6***) and Chrysosporazine P (***9***) 

Aliquots of **6** and **9** (0.1 mg, each) in 1 M HCl (0.3 mL) were heated to 100 °C, and the reaction was monitored by UPLC-DAD and UPLC-QTOF analyses of aliquots (50 µL) taken at 12, 24 and 36 h intervals and compared to the *N*,*N*-dideacylated product (**16**) obtained from a comparable acid hydrolysis of chrysosporazine D (**14**), as previously described [4]. (Appendix A, Appendix A)

### 3.8. Antibacterial Assay

The bacterium to be tested was streaked onto a tryptic soy agar plate and was incubated at 37 °C for 24 h. One colony was then transferred to fresh tryptic soy broth (15 mL) and the cell density was adjusted to 10^4^–10^5^ CFU/mL. The compounds to be tested were dissolved in DMSO and diluted with H_2_O to give 600 µM stock solution (20% DMSO), which was serially diluted with 20% DMSO to give concentrations from 600 µM to 0.2 µM in 20% DMSO. An aliquot (10 µL) of each dilution was transferred to a 96-well microtiter plate and freshly prepared microbial broth (190 µL) was added to each well to give final concentrations of 30–0.01 µM in 1% DMSO. The plates were incubated at 37 °C for 24 h and the optical density of each well was measured spectrophotometrically at 600 nm using POLARstar Omega plate (BMG LABTECH, Offenburg, Germany). Each test compound was screened against the Gram-negative bacteria *Escherichia coli* ATCC11775 and the Gram-positive clinical isolate bacteria methicillin-resistant *Staphylococcus aureus* and *Bacillus subtilis* ATCC 6633. Rifampicin, ampicillin and methicillin were used as a positive control (30 µM in 1% DMSO). The IC_50_ value was calculated as the concentration of the compound or antibiotic required for 50% inhibition of the bacterial cells using Prism 7.0 (GraphPad Software Inc., La Jolla, CA, USA) (Appendix A).

### 3.9. Antifungal Assay

The fungus *Candida albicans* ATCC 10231 was streaked onto a Sabouraud agar plate and was incubated at 37 °C for 48 h. One colony was then transferred to fresh Sabouraud broth (15 mL) and the cell density adjusted to 10^4^–10^5^ CFU/mL. Test compounds were dissolved in DMSO and diluted with H_2_O to give a 600 µM stock solution (20% DMSO), which was serially diluted with 20% DMSO to give concentrations from 600 µM to 0.2 µM in 20% DMSO. An aliquot (10 µL) of each dilution was transferred to a 96-well microtiter plate and freshly prepared fungal broth (190 µL) was added to each well to give final concentrations of 30–0.01 µM in 1% DMSO. The plates were incubated at 37 °C for 24 h and the optical density of each well was measured spectrophotometrically at 600 nm using a POLARstar Omega plate (BMG LABTECH, Offenburg, Germany). Ketoconazole was used as a positive control (30 µg/ml in 10% DMSO). Where relevant, IC_50_ values were calculated as the concentration of the compound or antifungal drug required for 50% inhibition of the fungal cells using Prism 7.0 (GraphPad Software Inc., La Jolla, CA, USA) (Appendix A).

### 3.10. Cytotoxicity Assay

Adherent SW620 (doxorubicin-susceptible human colorectal carcinoma) cells were cultured in Roswell Park Memorial Institute (RPMI) (New York, USA) 1640 medium. All cells were cultured as adherent monolayers in flasks supplemented with 10% foetal bovine serum, L–glutamine (2 mM), penicillin (100 unit/mL) and streptomycin (100 µg/mL), in a humidified 37 °C incubator supplied with 5% CO_2_. Briefly, cells were harvested with trypsin and dispensed into 96-well microtiter assay plates at 8000 cells/well, after which they were incubated for 48 h at 37 °C with 5% CO_2_ (to allow cells to attach as adherent monolayers). Test compounds were dissolved in 20% DMSO in PBS (*v*/*v*) and aliquots (10 µL) applied to cells over a series of final concentrations ranging from 10 nM to 30 µM. After 48 h incubation at 37 °C with 5% CO_2_ an aliquot (10 µL) of 3-(4,5-dimethylthiazol-2-yl)-2,5-diphenyltetrazolium bromide (MTT) in phosphate-buffered saline (PBS, 5 mg/mL) was added to each well (final concentration 0.5 mg/mL), and microtiter plates were incubated for a further 4 h at 37 °C with 5% CO_2_. After final incubation, the medium was aspirated and precipitated formazan crystals dissolved in DMSO (100 µL/well). The absorbance of each well was measured at 600 nm with a POLARstar Omega plate (BMG LABTECH, Offenburg, Germany). Where relevant, IC_50_ values were calculated using Prism 9.0 GraphPad Software, as the concentration of analyte required for 50% inhibition of cancer cell growth (compared to negative controls). Negative control was 1% aqueous DMSO, while positive control was vinblastine (30 µM). All experiments were performed in duplicate from two independent cultures. (Table 5, Figure 9)

### 3.11. MDR Reversal (Doxorubicin) Assay (P-glycoprotein Inhibition Assay)

The assay is similar to the above cytotoxicity (MTT) assay. However, instead of measuring the cytotoxicity of azachrysposorazines compounds, this assay was applied to measure the cytotoxicity of doxorubicin against multidrug-resistant SW620 Ad300 (P-gp-overexpressing human colorectal carcinoma) cells, in the presence and absence of PB42 compounds at concentrations that were non-cytotoxic to SW620 Ad300. SW620 Ad300 was cultivated in flasks as adherent monolayers in RPMI medium supplemented with 10% foetal bovine serum, L–glutamine (2 mM), penicillin (100 unit/mL), streptomycin (100 µg/mL) and doxorubicin (300 ng/mL) in a humidified 37 °C incubator supplied with 5% CO_2_. The cells were passaged 5 times and were maintained in 300 ng/mL of doxorubicin. On the day of the experiment, SW620 Ad300 cells were harvested with trypsin and dispensed into 96-well microtiter assay plates at 8000 cells/well in 180 µL medium per well, after which they were incubated for 48 h at 37 °C with 5% CO_2_. Following incubation for 48 h, azachrysposorazines A1 (**1**), A2 (**2**), B1 (**3**), C1 (**4**), C2 (**5**) and D1 (**6**), chrysosporazines N–P (**7**–**9**) and spirochrysosporazine A (**10**), chrysosporazines A–D (**11**–**14**) and chrysosporazine Q (**15**) (2.5 µM) were added to the wells containing a series of doxorubicin (30 – 0.01 µM). After 48 h incubation at 37 °C with 5% CO_2_, a solution of 3-(4,5-dimethylthiazol-2-yl)-2,5-diphenyltetrazolium bromide (MTT) in phosphate-buffered saline (PBS, 5 mg/mL) was added to each well (final concentration 0.5 mg/mL), and microtiter plates were incubated for a further 4 h at 37 °C with 5% CO_2_. After the media was carefully aspirated, the precipitated formazan crystals were dissolved in DMSO (100 µL). The absorbance of each well was measured at 600 nm with a POLARstar Omega plate (BMG LABTECH, Offenburg, Germany). Verapamil (2.5 µM) and DMSO served as positive and negative controls, respectively. All experiments were performed in duplicate from two independent cultures (Table 5, Figure 9).

## Figures and Tables

**Figure 1 marinedrugs-19-00478-f001:**
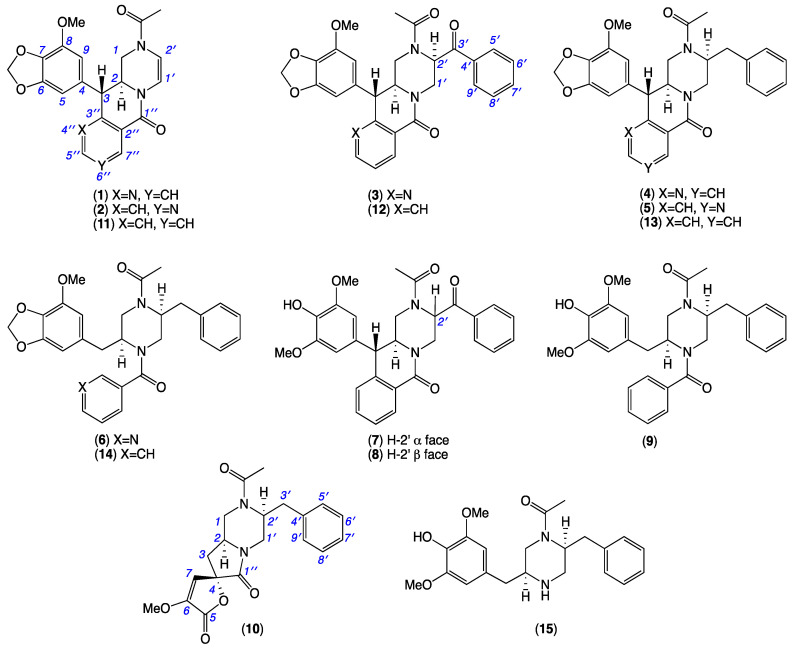
Metabolites **1**–**15** from *Chrysosporium* sp. CMB-F214.

**Figure 2 marinedrugs-19-00478-f002:**
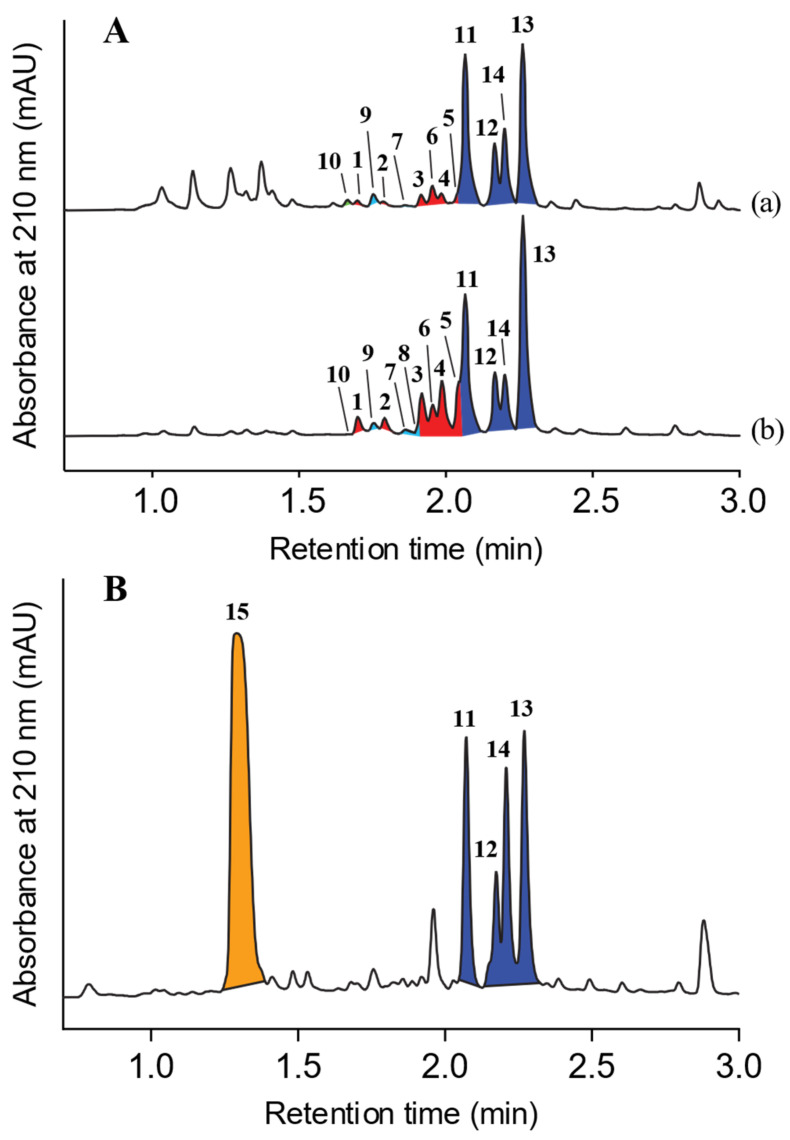
UPLC-DAD (210 nm) chromatograms of EtOAc extracts of *Chrysosporium* sp. CMB-F214 cultured in (**A**) M1 agar media (a) without addition, (b) with addition of 2 mg/mL sodium nicotinate; and (**B**) M2 agar media; peaks highlighted are new azachrysosporazines **1**–**6** (red), new chrysosporazines **7**–**9** (light blue) and spirochrysosporazine **10** (green), known chrysosporazines **11**–**14** (blue) and new chrysosporazine **15** (orange).

**Figure 3 marinedrugs-19-00478-f003:**
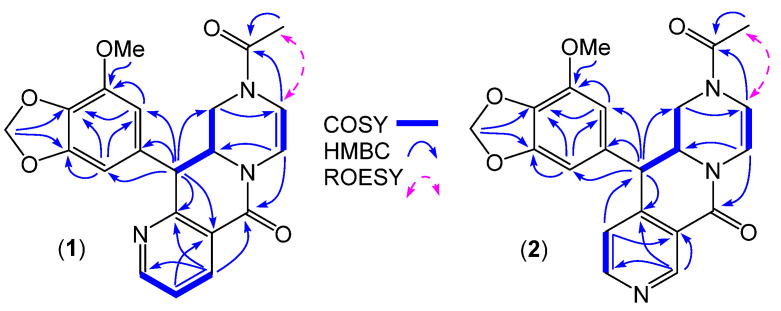
Selected 2D NMR (DMSO-*d*_6_) correlations for **1** and **2**.

**Figure 4 marinedrugs-19-00478-f004:**
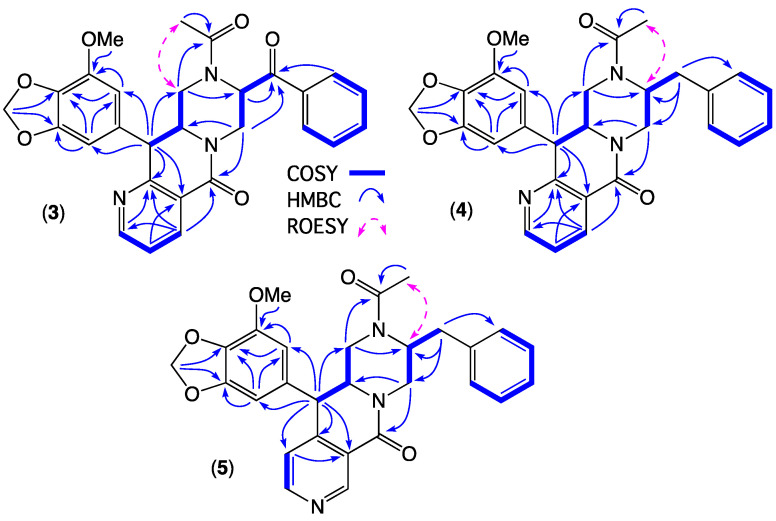
Selected 2D NMR (DMSO-*d*_6_) correlations for **3**–**5**.

**Figure 5 marinedrugs-19-00478-f005:**
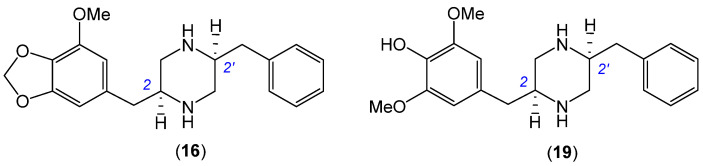
Piperazines **16** from acid hydrolysis of **6** and **14**, and **19** from acid hydrolysis of **9** and **15**.

**Figure 6 marinedrugs-19-00478-f006:**
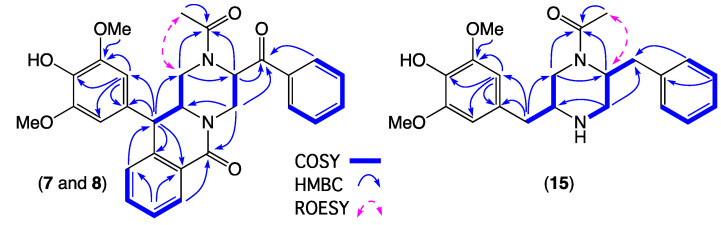
Selected 2D NMR (DMSO-*d*_6_) correlations for **7**–**8** and **15**.

**Figure 7 marinedrugs-19-00478-f007:**
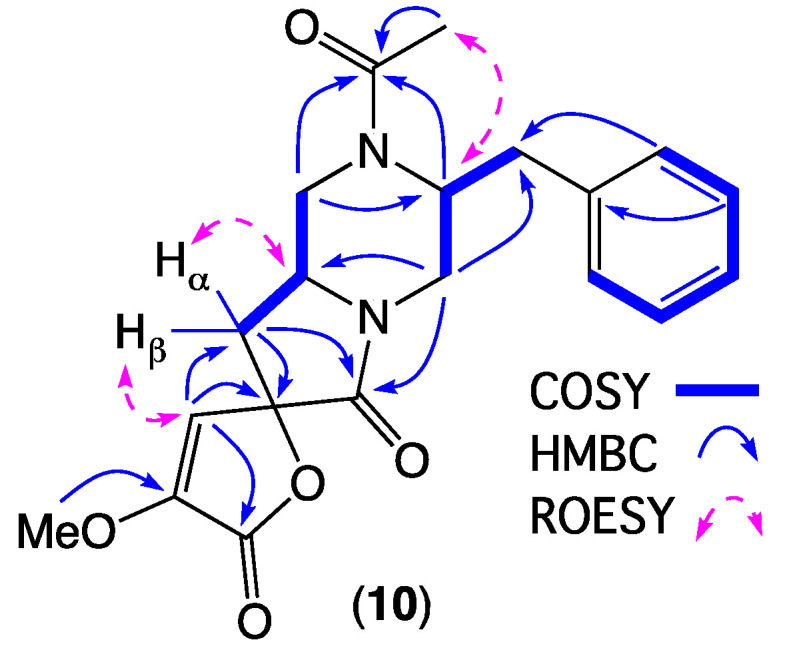
Selected 2D NMR (DMSO-*d*_6_) correlations for **10**.

**Figure 8 marinedrugs-19-00478-f008:**
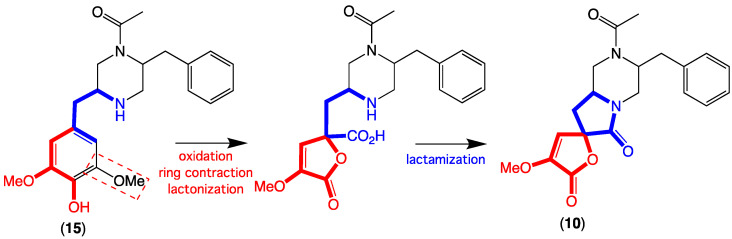
A plausible biosynthesis of **10** from **15**.

**Figure 9 marinedrugs-19-00478-f009:**
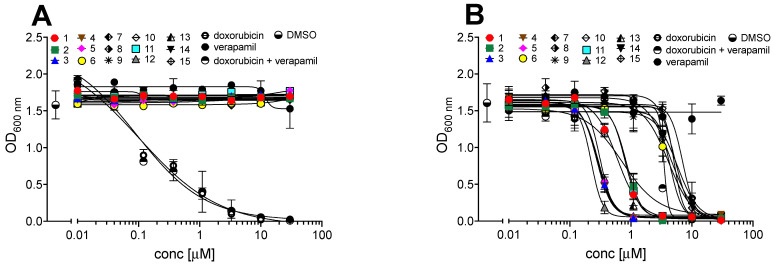
(**A**) Cytotoxicity of **1**–**15**, doxorubicin and verapamil against human colon (SW620) carcinoma cells. (**B**) Effect of **1**–**15** or verapamil (2.5 µM) on the sensitivity of P-gp-overexpressing human colon (SW620 Ad300) carcinoma cells to doxorubicin.

**Table 1 marinedrugs-19-00478-t001:** ^1^H NMR (DMSO-*d*_6_) data for compounds **1**–**5**.

Position	δ_H_, mult (*J* in Hz)	(2) δ_H_, mult (*J* in Hz)	(3) δ_H_, mult (*J* in Hz)	(4) δ_H_, mult (*J* in Hz)	(5) δ_H_, mult (*J* in Hz)
1	a. 4.26, ddd (*13.3,3.0, 1.2*)	a. 4.21, ddd (*13.1,3.0, 1.2*)	a. 3.98, dd (*12.9, 3.8*)	a. 4.29, dd (*13.8, 3.8*)	a. 4.19, dd (*13.8, 3.9*)
	b. 2.99, dd (*13.3, 9.8*)	b. 2.97, dd (*13.1, 9.8*)	b. 3.51 ^b^	b. 3.04 ^a^, dd (*13.8, 11.4*)	b. 2.99, dd (*13.8, 11.0*)
2	4.47, ddd (*12.3, 9.8, 3.0*)	4.49, ddd (*12.1, 9.8, 3.0*)	4.21, ddd (*9.8, 5.4, 3.8*)	3.88, ddd (*11.4, 10.0, 3.8*)	3.91, ddd (*11.2, 11.0, 3.9*)
3	4.51, d (*12.3)*	4.46, d (*12.1*)	4.46, d (*5.4*)	4.49, d (*10.0*)	4.46, d (*11.2*)
5	6.57, d (*1.4*)	6.68, d (*0.9*)	6.31, d (*1.4*)	6.49, d (*1.4*)	6.64, d (*1.3)*
9	6.65, d (*1.4*)	6.74, d (*0.9*)	6.55, d (*1.4*)	6.63, d (*1.4*)	6.73, d (*1.3)*
1′	6.80, d (*6.8*)	6.81, d (*6.8*)	a. 4.79, dd (*14.1*, *1.3*)	a. 4.56, dd (*13.4*, *1.1*)	a. 4.56, dd (*13.5, 1.2*)
	-	-	b. 3.51 ^b^	b. 3.00, m	b. 2.95, dd (*13.5, 4.1*)
2′	6.55, dd (*6.8, 1.2*)	6.56, dd (*6.8, 1.2*)	5.89, dd (*5.7, 1.3*)	4.25, m	4.27, m
3′	-	-	-	a. 3.03 ^a^, m	a. 3.06, dd (*13.4, 8.8*)
	-	-	-	b. 2.90, dd (*13.4, 5.8*)	b. 2.93, dd (*13.4, 5.8*)
5′/9′	-	-	7.97, m	7.26, m	7.26, m
6′/8′	-	-	7.56, m	7.30, m	7.30, m
7′	-	-	7.68, m	7.22, m	7.23, m
4″	-	6.66, d (*5.0*)	-	-	6.72, br s
5′′	8.56, dd (*4.8, 1.8*)	8.59, d (*5.0*)	8.64, dd (*4.7, 1.8*)	8.60, dd (*4.7, 1.8*)	8.60, br s
6′′	7.43, ddd (*7.8, 4.8, 0.8*)	-	7.44, dd (*7.8, 4.7*)	7.45, dd (*7.8, 4.7*)	-
7′′	8.26, dd (*7.8, 1.8*)	9.03, s	8.18, dd (*7.8, 1.8*)	8.18, dd (*7.8, 1.8*)	9.08, br s
NCOCH_3_	2.10, s	2.10, s	2.00, s	1.60, s	1.59, s
6-OCH_2_	6.02/6.00, Ab_q_	6.06/6.05, Ab_q_	5.95/5.94, Ab_q_	5.99/5.99, Ab_q_	6.05, br s
8-OCH_3_	3.80, s	3.81, s	3.78, s	3.80, s	3.82, s

^a^ overlapping resonances, ^b^ obscured by solvent signal.

**Table 2 marinedrugs-19-00478-t002:** ^13^C NMR (DMSO-*d*_6_) data for compounds **1**–**5**.

Position	(1) δ_C,_ Type	(2) δ_C,_ Type	(3) δ_C,_ Type	(4) δ_C,_ Type	(5) δ_C,_ Type
1	42.3, CH_2_	42.1, CH_2_	47.3, CH_2_	40.1 ^c^, CH_2_	39.8 ^c^, CH_2_
2	56.9, CH	56.5 ^a^, CH	59.2, CH	58.0, CH	56.9, CH
3	48.8, CH	45.4, CH	47.2, CH	48.8, CH	45.7, CH
4	132.8, C	131.1, C	135.3, C	134.6, C	132.6, C
5	103.4, CH	103.0 ^b^, CH	102.0, CH	103.1, CH	103.0, CH
6	148.5, C	149.0, C	148.6, C	148.5, C	148.9, C
7	134.0, C	134.6, C	133.8, C	133.8, C	134.4, C
8	143.1, C	143.6, C	143.1, C	143.1, C	143.5, C
9	110.1, CH	109.6 ^b^, CH	108.6, CH	109.6, CH	109.5, CH
1′	106.7, CH	106.4, CH	42.6, CH_2_	45.0, CH_2_	44.5, CH_2_
2′	112.6, CH	112.7, CH	54.3, CH	54.6, CH	54.3, CH
3′	-	-	197.6, C	34.9, CH_2_	34.8, CH_2_
4′	-	-	134.7, C	138.2, C	138.2, C
5′/9′	-	-	128.2, CH	129.4, CH	129.4, CH
6′/8′	-	-	128.9, CH	128.4, CH	128.4, CH
7′	-	-	133.5, CH	126.5, CH	126.5, CH
1″	158.5, C	157.4, C	161.9, C	163.5, C	162.9, C
2″	123.2, C	123.1, C	122.8, C	122.9, C	123.0, C
3″	158.6, C	149.8, C	158.1, C	158.4, C	149.3, C
4″	-	121.5, CH	-	-	121.5, CH
5″	152.4, CH	152.5, CH	153.0, CH	152.5, CH	152.4, CH
6″	122.9, CH	-	123.1, CH	122.7, CH	-
7″	135.4, CH	147.9, CH	135.6, CH	135.5, CH	148.3, CH
1-NCOCH_3_	166.7, C	166.7, C	170.5, C	168.4, C	168.3, C
1-NCOCH_3_	20.8, CH_3_	20.8, CH_3_	21.4, CH_3_	20.7, CH_3_	20.6, CH_3_
6-OCH_2_	101.2, CH_2_	101.6, CH_2_	101.3, CH_2_	101.2, CH_2_	101.5, CH_2_
8-OCH_3_	56.4, CH_3_	56.5 ^a^, CH_3_	56.4, CH_3_	56.4, CH_3_	56.4, CH_3_

^a^ assignments with the same superscript within a column are interchangeable, ^b^ detected from HMBC correlations, ^c^ obscured by solvent signal.

**Table 3 marinedrugs-19-00478-t003:** ^1^H NMR (DMSO-*d*_6_) data for compounds **7**–**8, 10** and **15**.

Position	(7) δ_H_, mult (*J* in Hz)	(8) δ_H_, mult (*J* in Hz)	(10) δ_H_, mult (*J* in Hz)	(15) δ_H_, mult (*J* in Hz)
1	a. 3.77, dd (13.6, 4.1)	a. 3.68, dd (14.3, 4.1)	a. 4.61, dd (13.3, 4.3)	a. 4.48, dd (14.5, 3.2)
	b. 3.43 ^a^	b. 3.61, dd (14.3, 6.8)	b. 2.77 ^b^, dd (13.3, 11.7)	b. 2.92 ^b^, m
2	4.27, ddd (12.4, 9.0, 4.1)	4.47, ddd (12.4, 6.8, 4.1)	3.60, m	3.27, m
3	4.36, d (9.0)	4.25, d (12.4)	α. 2.51 ^a^, dd (14.8, 7.2)	a. 2.88, dd (13.0, 6.2)
	-	-	β. 2.39, dd (14.8, 6.3)	b. 2.85, dd (13.0, 8.0)
5	6.54, s	6.61, s	-	6.55, br s
7	-	-	6.65, s	-
9	6.54, s	6.61, s	-	6.55, br s
1′	a. 4.76, dd(14.3, 1.9)	a. 4.37, dd (14.1, 7.1)	a. 3.79 ^c^, d (13.4)	a. 3.29, m
	b. 3.52, dd (14.3, 5.5)	b. 3.98, dd (14.1, 4.5)	b. 3.18, dd (13.4, 4.4)	b. 3.25, m
2′	5.83, dd (5.5, 1.9)	5.67, dd (7.1, 4.5)	4.21, m	4.30, m
3′	-	-	a. 2.93, dd (13.2, 6.9)	a. 3.21, dd (13.7, 10.4)
	-	-	b. 2.80 ^b^, dd (13.2, 8.5)	b. 2.93 ^b^, m
5′/9′	7.97, dd (8.1, 1.0)	8.03, dd (8.4, 1.3)	7.24 m	7.25 m
6′/8′	7.55, dd (8.1, 7.6)	7.58, dd (7.8, 7.6)	7.29, m	7.31, m
7′	7.67, t (7.6)	7.70, t (7.8)	7.23, m	7.25, m
4″	6.82, d (8.7)	6.65, d (7.8)	-	-
5′′	7.45, ddd (8.7, 7.4, 1.5)	7.42, ddd (7.8, 7.4, 1.3)	-	-
6′′	7.35, dd (7.7, 7.4)	7.35, dd (7.6, 7.4)	-	-
7′′	7.85, dd (7.7, 1.5)	7.85, dd (7.6, 1.3)	-	-
NCOCH_3_	1.91, s	1.83, s	1.69, s	1.46, s
6-OCH_3_	3.69, s	3.74, s	3.80 ^c^, s	3.76, s
8-OCH_3_	3.69, s	3.74, s	-	3.76, s
7-OH	8.43, br s	8.53, br s	-	8.03, br s
1′-NH	-	-	-	9.06, m

^a^ obscured by solvent signal, ^b,c^ assignments with the same superscript within a column are overlapping resonances.

**Table 4 marinedrugs-19-00478-t004:** ^13^C NMR (DMSO-*d*_6_) data for compounds **7**–**8, 10** and **15**.

Position	(7) δ_C,_ Type	(8) δ_C,_ Type	(10) δ_C,_ Type	(15) δ_C,_ Type
1	46.0, CH_2_	44.6, CH_2_	41.1, CH_2_	37.1, CH_2_
2	58.5, CH	57.7, CH	51.1, CH	55.9, CH
3	45.4, CH	46.9, CH	32.6, CH_2_	36.1, CH_2_
4	130.1, C	128.0, C	85.0, C	125.1, C
5	106.3, CH	106.6, CH	166.1, C	106.6, CH
6	148.3, C	148.5, C	147.0, C	148.0, C
7	134.7 ^a^, C	135.0, C	117.9, CH	134.6, C
8	148.3, C	148.5, C	-	148.0, C
9	106.3, CH	106.6, CH	-	106.6, CH
1′	40.9, CH_2_	38.8, CH_2_	43.0, CH_2_	45.9, CH_2_
2′	55.1, CH	57.1, CH	53.9, CH	52.3, CH
3′	196.6, C	196.3, C	34.7, CH_2_	34.7, CH_2_
4′	134.7 ^a^, C	134.6, C	137.9, C	137.6, C
5′/9′	128.2, CH	128.2, CH	129.4, CH	129.3, CH
6′/8′	128.9, CH	129.1, CH	128.4, CH	128.5, CH
7′	133.5, CH	133.8, CH	126.6, CH	126.8, CH
1″	162.8, C	162.5, C	166.7, C	-
2″	127.4, C	127.5 ^a^, C	-	-
3″	140.9, C	142.0, C	-	-
4″	127.7, CH	127.1 ^b^, CH	-	-
5″	132.4, CH	132.3, CH	-	-
6″	127.0, CH	127.1 ^b^, CH	-	-
7″	127.5, CH	127.5 ^a^, CH	-	-
1-NCOCH_3_	169.9, C	169.1, C	168.7, C	168.4, C
1-NCOCH_3_	21.2, CH_3_	20.9, CH_3_	21.0, CH_3_	20.3, CH_3_
6-OCH_3_	56.1, CH_3_	56.2, CH_3_	58.4, CH_3_	55.9, CH_3_
8-OCH_3_	56.1, CH_3_	56.2, CH_3_	-	55.9, CH_3_

^a,b^ assignments with the same superscript within a column are interchangeable.

**Table 5 marinedrugs-19-00478-t005:** Effect of metabolites **1–15** on inhibition of P-gp-mediated resistance to doxorubicin in human colon (SW620 Ad300) carcinoma cells, and cytotoxicity against doxorubicin-susceptible human colon (SW620) carcinoma cells.

SW620 Ad300				SW620	
Treatment	IC_50_ ^a^ (µM)	FR ^b^	GS ^c^	Treatment	IC_50_ ^a^ (µM)
Doxorubicin	5.75	57.5	1.0	Doxorubicin	0.10
+**1** (2.5 µM)	0.60	6.0	9.5	**1**	>30
+**2** (2.5 µM)	0.84	8.4	6.8	**2**	>30
+**3** (2.5 µM)	0.28	2.8	20.5	**3**	>30
+**4** (2.5 µM)	0.27	2.7	21.3	**4**	>30
+**5** (2.5 µM)	0.29	2.9	19.8	**5**	>30
+**6** (2.5 µM)	3.55	35.5	1.62	**6**	>30
+**7** (2.5 µM)	7.05	70.5	0.81	**7**	>30
+**8** (2.5 µM)	5.15	51.5	1.11	**8**	>30
+**9** (2.5 µM)	5.08	50.8	1.13	**9**	>30
+**10** (2.5 µM)	4.13	41.3	1.39	**10**	>30
+**11** (2.5 µM)	0.80	8.0	7.18	**11**	>30
+**12** (2.5 µM)	0.22	2.2	26.1	**12**	>30
+**13** (2.5 µM)	0.31	3.0	18.5	**13**	>30
+**14** (2.5 µM)	4.36	43.6	1.32	**14**	>30
+**15** (2.5 µM)	4.50	45.0	1.27	**15**	>30
+verapamil (2.5 µM)	0.71	7.0	8.1	doxorubicin + verapamil	0.092
verapamil	>30	--	--	verapamil	>30

^a^ MTT assay showing data as means of SEM of two independent cultures. ^b^ FR: fold resistance was determined by dividing the IC50 value for doxorubicin for P-gp-overexpressing cancer cells by the IC50 value for doxorubicin for sensitive cancer cells. ^c^ GS: Gain in sensitivity was the ratio of IC50 value of doxorubicin against SW620 Ad300 without testing compound to IC50 value of doxorubicin against SW620 Ad300 with testing compound. --: not calculated.

## Data Availability

Data is contained within the article or Appendix A.

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
