# Peer review of "Precursor-Directed Biosynthesis Mediated Amplification of Minor Aza Phenylpropanoid Piperazines in an Australian Marine Fish-Gut-Derived Fungus, Chrysosporium sp. CMB-F214"

_marinedrugs, 2021, doi:10.3390/md19090478_

Round 1

Reviewer 1 Report

 This is an excellent manuscript and has a good supplementary data in this manuscript. The paper is very well written and is clear and concise and easy to follow. The characterisation and structure elucidation of the isolated compounds is of a very high standard and the 2D NMR work in particular is to be commended. Overall the manuscript can be published in its present form.

Author Response

We thank the reviewer for the comment.

Reviewer 2 Report

The manuscript describes the isolation of new chrysosporazine derivatives, of which the production by a marine fish-gut derived fungus was enhanced by the addition of sodium nicotinate. The experiments were designed and carried out appropriately, and the structure elucidation of novel analogs has been provided based on unambiguous interpretation of spectroscopic data. The depiction of the results (Tables and Figures) is very neat and understandable. Overall, this manuscript is of high quality and has significance in the field of natural product chemistry. A few minor concerns are as below.

  • Figure 5 is not mentioned in the main text.
  • The authors suggest the biosynthesis pathway for 10 in figure 8, without detailed description in the main text. It would be very helpful for readers if more explanation (with reference, if possible) could be provided. Would the cleavage of the aromatic ring proceed in a manner similar to that during the biosynthesis of jadomycin?  

Author Response

Reviewer 2: Publish after minor revisions 

1. Overall, this manuscript is of high quality and has significance in the field of natural product chemistry. A few minor concerns are as below. Figure 5 is not mentioned in the main text.

Figure 5 has been added to the text, Line 163 and the manuscript has been updated accordingly.

2. The authors suggest the biosynthesis pathway for 10 in figure 8, without detailed description in the main text. It would be very helpful for readers if more explanation (with reference, if possible) could be provided. Would the cleavage of the aromatic ring proceed in a manner similar to that during the biosynthesis of jadomycin?

We believe that the provided explanation is sufficient as this is only a suggested biosynthetic pathway. Without a more detailed study on the gene cluster, we will not know if it proceeds in the same manner as jadomycin. No action taken

Reviewer 3 Report

The authors analyzed the chemical profile of fish-gut derived fungus on different culture conditions with MATRIX method.  They found the culture condition to improve productivity of new compounds. Totally the authors discovered 11 new compounds along with four previously discovered compound. The general structure elucidation strategy is strongly supported by their previous report. Also, they found very potent activity of new compounds. The manuscript is describing very interesting research findings and new approaches. Therefore, it would be of interest to the readers of Marine Drugs. Furthermore, the topic is suitable to the special issue. However, there is several issues listed below and it should be clarified before acceptance.

Table 1: The multiplicity of 6-OCH2 in 1 was described as dd. However, there is no partner in the molecule to generate the coupling of 11.2 and 0.9. The authors need to analyze the multiplicity of 6-OCH2 proton carefully. Furthermore, some of coupling need to be measured precisely. (JH-1,H-2’=1.3 Hz; JH-2’,H-1=1.2 Hz) Similar issues were found in the other compounds data. Please inspect the spectra again.

The vicinal coupling between H-2 and H-3 of compound 3 looks much smaller than those of compounds 1, 2, 4, 5 and 12. Compounds 7 and 8 share the core piperazine structure with compound 3, but the vicinal coupling constant between H-2 and H-3 looks quite different. Therefore, it needs careful inspection of spectroscopic data or other reliable method.

Although H-3a was indicated to be obscured by solvent signal, the authors measured coupling constants. If the authors performed additional experiments to measure coupling constants, please explain.

HMBC spectra are critical evidences of the structure elucidation. However, any HMBC spectra is not included in the supplementary materials. Please add the HMBC spectra in the Supplementary Materials

In Figure 8, the authors used “n” as superscripts. It would be recommended to use full description.

Format

Please relocate reference number throughout the manuscript.

           CMB-F585;[1] -> CMB-F585 [1];

           respectively.[4,5] -> respectively [4,5].

Line 377: “-“ -> “–“

Line 391: “L” in L-glutamine need to be small-capitalized.

Table 3: “H” need to be as a subscript. “J” need to be italicized.

Author Response

Reviewer 3: Publish after minor revisions

1. Table 1: The multiplicity of 6-OCH2 in 1 was described as dd. However, there is no partner in the molecule to generate the coupling of 11.2 and 0.9. The authors need to analyze the multiplicity of 6-OCH2 proton carefully. Furthermore, some of coupling need to be measured precisely. (JH-1,H-2’=1.3 Hz; JH-2’,H-1=1.2 Hz) Similar issues were found in the other compounds data. Please inspect the spectra again.

We agree with the reviewer that OCH2 should not be dd. However, this dioxymethylene OCH2 signal is a typical example of an ABq spin system that consists of 4 peaks and an indicative of JAB system. ABq is common for molecules that contain two isolated protons coupled only to each other. Consequently. we have revised the J value for 6-OCH2 and changed it to doublet for compounds 1 – 4.

In addition, we have revised and fixed the J values for the other compounds and amended the manuscript and supporting information accordingly.

2. The vicinal coupling between H-2 and H-3 of compound 3 looks much smaller than those of compounds 1, 2, 4, 5 and 12. Compounds 7 and 8 share the core piperazine structure with compound 3, but the vicinal coupling constant between H-2 and H-3 looks quite different. Therefore, it needs careful inspection of spectroscopic data or other reliable method.

We have rechecked the NMR data, and the coupling constants remain the same. This could be due to the special orientation of the compound 3 (as it’s major rotamer is different from other chrysosporazines, same applies to the compound 7). No action taken

3. Although H-3a was indicated to be obscured by solvent signal, the authors measured coupling constants. If the authors performed additional experiments to measure coupling constants, please explain.

For compound 10, although part H-3a signal was obscured by DMSO, we can still observe the other half that allowed us to assign the J value. No action taken

4. HMBC spectra are critical evidence of the structure elucidation. However, any HMBC spectra is not included in the supplementary materials. Please add the HMBC spectra in the Supplementary Materials

The HMBC spectra have been added to the supporting information. Therefore, the manuscript and the supporting information have been updated accordingly.

5. In Figure 8, the authors used “n” as superscripts. It would be recommended to use full description.

Full description has been added to the manuscript.

6. Formatting Requests:

Please relocate reference number throughout the manuscript.

CMB-F585;[1] -> CMB-F585 [1];

respectively.[4,5] -> respectively [4,5].

Line 377: “-“ -> “–“

Line 391: “L” in L-glutamine need to be small-capitalized.

Table 3: “H” need to be as a subscript. “J” need to be italicized

All formatting requests have been addressed and the manuscript has been amended accordingly.

Round 2

Reviewer 3 Report

The authors clarified all the issues raised by reviewers. Therefore manuscript could be accepted.

One last suggestion is adding the explanation on the rotamer issue related to the vicinal coupling constants between H-2 and H-3 of compound 3 in the manuscript. It will help the readers understand the data clearly. 

Author Response

We thank the reviewer for the suggestion. While there are clearly conformational changes that influence the J2,3 vicinal coupling, allowing it to range from ~5 to >12 Hz across different co-metabolites, we can see no clear trend linking J2,3 to either E or Z acetamide rotamers. For example, where (3) and (7-8) exhibit a Z acetamide as the major rotamer, their J2,3 are 5.4, 9.0 and 12.4 Hz, respectively. Likewise, where (4-5) both feature an E acetamide as the major rotamer, their J2,3 values are 10.0 ad 11.2 Hz respectively. No action taken